# The Oral Inactivated Porcine Epidemic Diarrhea Virus Presenting in the Intestine Induces Mucosal Immunity in Mice with Alginate–Chitosan Microcapsules

**DOI:** 10.3390/ani13050889

**Published:** 2023-02-28

**Authors:** Ziliang Qin, Zida Nai, Gang Li, Xinmiao He, Wentao Wang, Jiqiao Xia, Wang Chao, Lu Li, Xinpeng Jiang, Di Liu

**Affiliations:** 1College of Animal Science and Technology, Northeast Agricultural University, Harbin 150030, China; 2Key Laboratory of Combining Farming and Animal Husbandry, Ministry of Agriculture, Animal Husbandry Research Institute, Heilongjiang Academy of Agricultural Sciences, No. 368 Xuefu Road, Harbin 150086, China

**Keywords:** microcapsules, PEDV, mucosal immunity, alginate, chitosan

## Abstract

**Simple Summary:**

Porcine epidemic diarrhea virus (PEDV) is an α coronavirus that causes major disease outbreaks, producing up to 100% mortality rates in piglets during the first 7 days after birth. In this study, we used microcapsules with inactivated PEDV fed to mice by oral administration to improve the effectiveness of the oral delivery method for protection against PEDV infection, and avoided digestive degradation in the acidic environment of the stomach. In addition, the PEDV microcapsules displayed remarkable storage tolerance to maintain the quality of the PEDV antigen. The PEDV microcapsules delivered the inactivated virus into the gut, stimulating the specific mucosal immune response in mice, which could directly neutralize the enterovirus.

**Abstract:**

The porcine epidemic diarrhea virus, PEDV, which causes diarrhea, vomiting and death in piglets, causes huge economic losses. Therefore, understanding how to induce mucosal immune responses in piglets is essential in the mechanism and application against PEDV infection with mucosal immunity. A method of treatment in our research was used to make an oral vaccine that packaged the inactive PEDV with microencapsulation, which consisted of sodium alginate and chitosan, and adapted the condition of the gut in mice. The in vitro release experiment of microcapsules showed that inactive PEDV was not only easily released in saline and acid solutions but also had an excellent storage tolerance, and was suitable for use as an oral vaccine. Interestingly, both experimental groups with different doses of inactive virus enhanced the secretion of specific antibodies in the serum and intestinal mucus, which caused the effective neutralization against PEDV in the Vero cell by both IgG and IgA, respectively. Moreover, the microencapsulation could stimulate the differentiation of CD11b+ and CD11c+ dendritic cells, which means that the microencapsulation was also identified as an oral adjuvant to help phagocytosis of dendritic cells in mice. Flow cytometry revealed that the B220^+^ and CD23^+^ of the B cells could significantly increase antibody production with the stimulation from the antigens’ PEDV groups, and the microencapsulation could also increase the cell viability of B cells, stimulating the secretion of antibodies such as IgG and IgA in mice. In addition, the microencapsulation promoted the expression of anti-inflammatory cytokines, such as IL-10 and TGF-β. Moreover, proinflammatory cytokines, such as IL-1, TNF-α, and IL-17, were inhibited by alginate and chitosan in the microencapsulation groups compared with the inactivated PEDV group. Taken together, our results demonstrate that the microparticle could play the role of mucosal adjuvant, and release inactivated PEDV in the gut, which can effectively stimulate mucosal and systemic immune responses in mice.

## 1. Introduction

Porcine epidemic diarrhea virus, PEDV, is a highly contagious disease that infects the small intestinal epithelial cells of piglets during the first seven days of birth, and the mortality rate can range from 70% to 100% in three-day-old piglets [1]. At the first week of birth, the major immune protection of piglets is dependent on maternal antibodies and innate immunity. The maternal antibody is induced by the inactivated and attenuated vaccines of PEDV through the intramuscular route or subcutaneous injection in the pregnant sow, but the maternal antibody is digested by gastric acid and pepsin before entering the intestine, which cannot effectively protect against PEDV infection. However, mucosal immunity can be built no more than three days after birth, and provides more effective protection than systemic immunity in piglets. Some studies have demonstrated that the first line of defense by IgA in the intestine would be more effective than that by IgG for neutralizing the infection of intestinal diseases [2]. However, the problem of how to transport the inactivated PEDV virus into the intestine and activate mucosal immunity has not yet been solved. Moreover, the mechanism of mucosal immunity to PEDV in the gut of fetal pigs remains unclear. The process of specialized M-cells taking up and presenting PEDV to the dendritic cells and lymphoid follicles also needs to be better understood to learn how antigens arrive at the inductive sites for T-cell- and B-cell-dependent activation of IgG and IgA production. One interesting strategy is packaging vaccines to look like microcapsules, which have different properties for oral immunity, such as size, shape, and surface molecule organization [3].

Chitosan and alginate are mixed to coagulate the ionic gel, which is a potential candidate vehicle for their main constituents due to their intrinsic immunomodulatory properties. Alginate is one of the most widely used carriers for the controlled release of different types of active agents. Moreover, sodium alginate, which is derived from marine brown algae and bacteria, has received attention as an antimicrobial in the food industry because of its unique physicochemical properties and biological activity. Chitosan (CS), also called poly glucosamine, is widely used to enhance the mucosal immune response due to its biocompatibility, biodegradability, and mucoadhesive properties [4]. The production method of alginate–chitosan microcapsules by using ionic gelation is an interesting approach to compound oral delivery systems for inactivated PEDV vaccines. The classic technique involves first mixing the antigen of inactivated PEDV and alginate solution and then pouring it slowly into a solution of calcium chloride and chitosan. Finally, the alginate microcapsules are collected by decanting the supernatant and are washed with water. Hence, the preparation method of PEDV alginate microcapsules is very simple. The manufacture of microencapsulations showed that they used simple machinery, had a low cost, and a good safety profile, making them an attractive approach to oral immunization. In a previous study of the mucosal immune system, oral vaccination with recombinant Lactobacillus was more promising for stimulating specific immunization against pathogens in the first contact with mucosal surfaces [5,6,7]. To improve the effectiveness of the oral vaccination delivery method, which protects against PEDV infection, avoiding digestive degradation in the acidic environment of the stomach is crucial (oral vaccination with alginate microsphere systems). There is also a novel microencapsulation method for rotavirus with anionic polymers and amines, including sodium alginate and spermine hydrochloride, which increases the antibody titers against rotavirus infection [8]. Similarly, microencapsulations of PEDV can arrive at the gut of piglets and release the virus in an alkaline environment. Moreover, they can be taken up by mucosa-associated lymphoid tissues to activate mucosal immune responses against encapsulated PEDV, and be processed by Peyer’s patches and M-cells in the small intestine [9].

In this study, inactivated PEDV packaged in microencapsulations was used to explore an efficient oral vaccine delivery system to release the virus into the gut, which would be better for protecting against PEDV-induced specific mucosal immunity in mice. In addition, mucosal and systemic immune responses were monitored for specific antibodies, such as IgA and IgG, in the feces and serum of mice, and then used to analyze the neutralizing activity against PEDV. Most importantly, the micro-encapsulations of PEDV were used with the antigen of complete enterovirus particles to present into the mucosal immune cells in the gut. The functional immune cells in the specific immune response, such as dendritic cells and B cells, were studied to focus on the mechanism of complete virus particles in the recognition, presentation, and antigen processing with oral mucosal immunity against enterovirus infection in vivo.

## 2. Materials and Methods

### 2.1. Cell and Virus

PEDV strain LJB/03 was previously isolated from diarrheic piglets in China and purified in our laboratory [10]. Vero cells were cultured in Dulbecco’s modified Eagle’s medium (DMEM, Gibco) supplemented with 10% fetal bovine serum (FBS, Gibco) at 37 °C with 5% CO_2_. A PEDV strain (LJB/03) was isolated and stored in our laboratory. Confluent Vero cells grown in a 5 mL tissue culture flask were washed with PBS three times and inoculated with 1 mL of a mixture solution containing trypsin at a concentration of 10 µg/mL, PEDV inoculation culture (3 × 10^4^ PFU), and DMEM without any serum. After incubating at 37 °C for 1 h, 4 mL of DMEM with trypsin at a concentration of 8 µg/mL was added. Before 70% cytopathic effects, the cells were maintained at 37 °C under 5% CO_2_ and monitored daily. Finally, the diseased cells were subjected to three rounds of freezing and thawing, and then the supernatants were collected after centrifugation for 10 min at 12,000 rpm. The virus titers were measured with the method of TCID50, and the PEDV titer was approximately 6.3 × 10^7^ PFU. PEDV was inactivated with 0.1 M binary ethylenimine (BEI) to a final volume of 5%, which was incubated at 37 °C for 24 h. In addition, sodium thiosulfate was used to neutralize the excess BEI at 37 °C for 2 h. Inactivated PEDV virus was stored at −80 °C until use.

### 2.2. The Production of Microcapsules

The inactivated PEDV (6 × 10^6^ PFU and 6 × 10^7^ PFU) and sodium alginate solution at 1.5% (wt/vol) were obtained by dissolving and passing the solution. The CaCl_2_-chitosan solution was obtained by adding CaCl_2_ and sodium alginate to final concentrations of 4% and 1% (wt/vol), respectively. After completely mixing the PEDV and the sodium alginate solution, the mixture was dropped at a constant speed (approximately 60–90 drops/min) through a medical 9-gauge syringe into a solution of calcium chloride and chitosan acetic acid. A suspension of sodium alginate–chitosan microcapsules was obtained after thorough stirring at a constant rate of approximately 300 r/min at room temperature for 30 min. The particle size was approximately 7.54 ± 2.87 μm, which was measured under a light microscope. Finally, the samples were separated, filtrated, and washed repeatedly with distilled water, precooled at −80 °C for 1 h, and freeze-dried.

### 2.3. Analysis of Encapsulation Efficiency and Protein Content

The 100 mg freeze-dried microcapsules were weighed into a 50 mL centrifuge tube with 10 mL trisodium citrate solution (0.06 mol/L) and incubated at room temperature for 10 min. The microcapsules were ultrasonicated and centrifuged at 10,000 rpm for 10 min. Finally, approximately 1 mL of the supernatant (filtrated processing if there was another matter) was collected to determine the total protein content by the Bradford Protein Assay Kit (Coomassie Brilliant Blue) [11]. The encapsulation efficiency (B) and the amount of carrier protein (Z) of the microcapsule were calculated according to the formula [12]:B = W/J × 100%(1)
Z = W/M(2)

(W: protein content in the microcapsules; J: total amount of protein; M: total amount of microcapsules.)

### 2.4. In Vitro Release Experiment of Microcapsules in Different Saline Solutions, pH Values, and Room Temperature Storage Tolerances

Three 50 mL centrifuge tubes containing 100 mg of PEDV microcapsules were added to 10 mL of pH 7.4 PBS solution, 0.75% physiological saline solution, and pH 2.3 hydrochloric acid solution, and then placed in the Oven-Controlled Crystal Oscillator at 37 °C and a speed of 100 r/m. The release of protein content in the solution was measured at 1 h, 3 h, 9 h, 1 d, 3 d, 6 d, 12 d, and 18 d. The protein content was analyzed by the Coomassie Brilliant Blue method [13], and then the release rate (RT) curve was drawn.

The PEDV microcapsules of freeze-dried particles were stored without any media in the dark at room temperature for 5 months, and the storage performance was evaluated by measuring the changes in the PEDV protein release rate every month. The concrete method of PEDV protein release rate refers to analysis of encapsulation efficiency and protein content. The release rate (RT) was calculated according to the formula: RT = R/W. (R: release amount of PEDV protein; W: the protein content in the microcapsule) [3].

### 2.5. Immunization of Microcapsules in SPF Mice

A total of 50 SPF mice, 5 weeks old, with approximately 18 g of body weight, were randomly divided into five groups. Each group contained 10 mice. The immunization regimen is shown in Table 1. The first group, the negative control group, was immunized with PBS buffer through oral administration. The second group was immunized with microcapsules without virus through oral administration. The third group was immunized with inactivated PEDV without encapsulation through oral administration. The fourth group was immunized with the encapsulated PEDV vaccine through oral administration, which encapsulated approximately 6 × 10^6^ PFU of PEDV. The fifth group was immunized with approximately 6 × 10^7^ PFU of encapsulated PEDV through oral administration, which encapsulated 100 μL of PEDV. All the groups were used to immunize mice via an intragastric route on Days 1 and 3 twice. On Days 0 (preimmunization), 3, 6, and 9, 200 mg serum samples and feces samples were collected from both the tail vein and the anus. The fecal samples were subsequently homogenized for 30 min in 400 μL of sterile PBS (pH 7.4) containing 0.01 mol/L EDTA-Na_2_ and then incubated for 12 h at 4 °C. The extracted supernatants of all fecal samples were collected by centrifugation at 15,000× *g* for 10 min and stored at −80 °C [6]. The feces were supplemented with protease inhibitors for the subsequent ELISA analysis. The serum was stored at −80 °C for the subsequent ELISA and neutralization analyses. Intestinal lavage samples (mucus) were obtained from the intestine of euthanized mice. The mucus was suspended in 400 μL of sterile PBS (pH 7.4) containing 0.01 mol/L EDTA-Na_2_ for 2 h at 4 °C. Then, all mucus samples were collected by centrifugation at 15,000× *g* for 10 min, and the supernatant was stored at −80 °C with protease inhibitors for neutralization analysis.

### 2.6. Enzyme-Linked Immunosorbent Assay (ELISA)

The ELISA plates were coated for 18 h at 4 °C with PEDV (6.3 × 10^7^ PFU), which was previously cultured from Vero cells in the previous section. After the plates were blocked with 5% skim milk in PBS for 1.5 h at 37 °C, they were washed 3 times with PBST (PBS + 0.1% Tween 20). Sera and fecal samples (supernate of fecal) were added into wells in triplicate and used to test specific antibodies such as IgG and IgA. The sera and fecal samples were incubated for 1 h at 37 °C and washed as before. Goat anti-mouse IgG and IgA antibodies–HRP (Invitrogen) were added to each well (1:5000) and incubated for 1 h at 37 °C. After the final round of washing with PBST three times, the TMB substrate was used for color development, and the absorbance was measured at 490 nm.

### 2.7. PEDV Neutralization Assay

The serum of mice fed with PEDV microcapsules and inactivated PEDV was collected to determine the neutralization ability of antibodies. The serum and intestinal lavage of mucus were obtained from the previously described immunization, and were filtered with a 0. 45 µm filter membrane. The concrete method was as follows: 50 µL samples (serum and intestinal lavage) were taken to a constant dilution from 1:2 to 1:512 and then added to a 96-well plate for eight replications with Vero cells. PEDV at a titer of 5 × 10^5^ PFU was mixed within the DMEM culture medium, and 10% heat-inactivated bovine serum albumin was added to the cell plate, which was coated with the diluted serum and fecal sample solution and then incubated with the antibody and virus at 37 °C for 1 h. Then, 100 µL of Vero cells were added to the antibody–virus mixture and incubated in a 5% CO_2_ incubator at 37 °C for 3 days. Finally, the covered medium was discarded, and the wells were washed three times with sterile PBS (pH = 7.4) and dyed with 1% crystal violet solution. The quantity variance of plaque was analyzed to demonstrate their antibody level.

### 2.8. T-Cell Proliferation

Five mice were dissected from each group on the seventh day after the second immunization. Single-lymphocyte suspensions were prepared from the spleen after the second immunization, as previously described [5]. Lymphocytes were isolated from the spleens of five mice dissected from each group and incubated in triplicate in 96-well plates at 5 × 10^5^ cells/well with Roswell Park Memorial Institute 1640 medium (RPMI-1640) plus 20% fetal calf serum (FCS) at 37 °C in a 5% CO_2_ incubator. Then, the cells were stimulated for 48 h with 0.5 and 1 μg/mL inactive PEDV in the experimental group (specific antigen stimulation) and control group (without antigen stimulation). Based on the manufacturer’s instructions (Promega), the Cell Titer 96 Aqueous Non-Radioactive Cell Proliferation Assay was used to evaluate T-cell proliferation. The solution of A thiazolyl blue (MTT) was pipetted (10 µL) into each well to develop the color. The plates were incubated for 4 h before reading the OD_490_ values of the plate using the Magellan plate reader, and then the average was taken of the PEDV stimulation data to compare with that of the negative control wells.

The proliferation rate was analyzed with the stimulation index: SI = (OD490 experiment group − OD490 culture media)/(OD490 control group − OD490 culture media)

### 2.9. Flow Cytometry and Cell Sorting

The suspensions of single lymphocytes were derived from the spleens of immunizing mice that were dissected from each group on the seventh day of the second immunization [14]. The isolated spleen T-cells (5 × 10^5^ cells/mL) were cultured in RPMI 1640 culture media for flow cytometry staining. The incubated single-lymphocyte suspensions were stained with anti-CD11c (PE-conjugated) and anti-CD11b (FITC-conjugated) antibodies and anti-CD23 (APC-conjugated) and anti-B220 (PE-conjugated) antibodies in RPMI 1640 culture media without serum. All antibodies were purchased from Miltenyi Biotec. Single lymphocyte cells were stained with CD11c (PE-conjugated) and CD11b (FITC-conjugated) antibodies and sorted with FACStar to prepare CD11c^+^ and CD11b^+^ cells. The same method was used to prepare CD23^+^ B220^+^ cells and B220^+^ cells. All the samples were tested with FACSCanto (BD Biosciences) and analyzed by CellQuest software [15]. In all cases, the purity of FACStar-sorted cells was >98%. The cells were sorted by FACSAria (BD Biosciences) into CD11c^+^, CD11b^+^, CD23^+^B220^+^, and B220^+^ populations in complete medium (RPMI, 1% penicillin-streptomycin, 1% glutamine, 50 mM b-mercaptoethanol).

### 2.10. Cytokine Assays

Real-time qRT-PCR was used to determine the levels of cytokine genes and intestinal tight junction gene products in splenic lymphocyte and intestinal samples from immunized mice using a CFX96TM Real-Time PCR Detection System (Bio-Rad, Hercules, CA, USA). Total RNA was extracted from intestinal tissues and spleen with total RNA extraction kits (TaKaRa, Dalian, China) according to the manufacturer’s instructions [16]. Total RNA was converted to cDNA using the PrimeScript^TM^ RT reagent Kit with gDNA Eraser (TaKaRa, Dalian, China) [17]. The cDNA products were used for real-time PCR with a SYBR Premix Ex Taq^TM^ II reagent kit (TaKaRa, Dalian, China), and the specific primers used are listed in the Appendix A [18]. The Livak method (2^−∆∆CT^ method) was used to calculate the fold change compared to β-actin gene controls [19,20].

### 2.11. Statistical Analysis

All the samples and experiments were analyzed in triplicate, and all the groups were examined as independent measurements to support adequate statistics in the experiments. The average duration of recovery and colonization over time were compared between groups using repeated measures analysis of variance with Bonferroni’s correction using SPSS software (IBM, New York, USA). Data between the different groups were analyzed with one-way repeated measures analysis of variance (ANOVA) and the least significant difference (LSD) test with GraphPad software. Statistically significant effects (*p* < 0.05) were further analyzed [21].

### 2.12. Ethical Statement

All animals were housed in negative-pressure isolators with HEPA filters in a BSL2. The protocols for animal experiments were approved by the Institutional Committee of Northeast Agricultural University (2018NEAU-131, 12 September 2018) and complied with the guidelines of the Northeast Agricultural University Administrative Committee of Laboratory Animals.

## 3. Results

### 3.1. The Excellent Storage Tolerances of Microcapsules in Different Saline Solutions, pH Values, and Room Temperature 

As a biomedical material, the ability of microcapsules to be released in saline and acid solutions is crucial in clinical applications. The release rates in different saline solutions, pH values, and storage times are shown in Figure 1. The trend of the in vitro release ability of microcapsules in the PBS group and normal saline group was similar, and the release rate was more than 50% on the 3rd day; on the 18^th^ day, the release rate reached 95% and 91%, respectively. However, that of the hydrochloric acid group was only 37.5% on the 3rd day and just 63.2% on the 18th day. These results indicated that the microcapsules were easily released in saline solution, but most importantly, they were acid-stable. The release ability after different months of storage showed that the release rate remained higher than 83% in the first three months, and they still maintained the lowest release rate of 41% in the fifteenth month. These results indicated that the microcapsules have excellent storage tolerance in the dark at room temperature for 5 months.

### 3.2. The Microcapsules Stimulate the Specific Humoral and Mucosal Immunity 

The specific antibodies in intestinal lavage and serum were detected by ELISA at the end of the immunization. The ELISA results showed that with increasing days, the specific antibody level in both the inactivated PEDV and microcapsule groups continuously increased, as shown in Figure 2. For specific IgA, the PEDV (low) and PEDV (high) microcapsule groups had significantly higher levels than the inactivated group on the third day (*p* < 0.05), and the PEDV (high) microcapsules group had remarkably higher levels than the PEDV (low) microcapsules group (*p* < 0.05). The PEDV (low) microcapsules group had significantly higher levels than the inactivated PEDV (*p* < 0.05) on the sixth day, while the PEDV (high) microcapsules group had significantly higher levels than the inactivated group (*p* < 0.01). At the end of the detection on the ninth day, both PEDV microcapsules groups had significantly higher levels than the inactivated group (*p* < 0.01). Most importantly, the empty microcapsule group did not have detectable IgA antibodies during the whole immunization period. Figure 2B shows that the specific IgG levels in both PEDV microcapsules showed the same trend as the intestinal lavage results and were significantly higher than those in the inactivated PEDV group on the third day (*p* < 0.05). The specific IgG levels in the PEDV microencapsulated groups were significantly higher than those in the inactivated PEDV group on the sixth and ninth days (*p* < 0.01). These results suggest that the PEDV microcapsules have the ability to activate specific mucosal and systemic immune responses in a dose-dependent manner in mucosal immunity.

### 3.3. The Specific IgG and IgA Neutralized PEDV

The neutralization of IgG (Figure 3A) and IgA (Figure 3B) is shown in Figure 3. The results indicated that the neutralization of IgG in the PEDV (high) microcapsule group was more effective than that in the PEDV (low) microcapsule group after the 1:8 dilution point against PEDV infection in the in vitro experiment. The 50% neutralization with IgG of the high PEDV group was probably 1:256, while the PEDV (low) microcapsule group was 1:128. The 50% neutralization of the PEDV (high) microcapsule group was higher than the PEDV (low) microcapsule group in the neutralization. The neutralization of IgA in both the PEDV (high) and PEDV (low) microcapsule groups was higher than that in the inactivated PEDV group, inhibiting PEDV infection in the in vitro experiment. The 50% neutralization of IgA for the inactivated PEDV group, PEDV (low) microcapsule group, and PEDV (high) microcapsule group gradually increased against PEDV infection in Vero cells at dilutions of 1:16, 1:32, and 1:64, respectively. The neutralization ability of IgG and IgA in the PEDV (high) microcapsule group was higher than that in the PEDV (low) microcapsule group. The neutralization of PEDV microcapsules was also dose-dependent, which indicated that the PEDV (high) microcapsule group had better systemic and mucosal immunity than the PEDV (low) microcapsule group.

### 3.4. PEDV Microcapsules Stimulated Immune Memory

To further analyze whether microcapsules influence cell-mediated immunity, T-cell proliferation was analyzed to study immune memory with PEDV from five immunized mice for 29 days after the third immunization to obtain a single-cell suspension of lymphocytes for the in vitro experiment (Table 2). The suspension was stimulated with PBS (as a control), microcapsules (without PEDV), inactivated PEDV, PEDV (low) microcapsules, and PEDV (high) microcapsules. Different doses of 0.5 μg/mL and 1.0 μg/mL PEDV were used to simultaneously stimulate lymphocyte proliferation. At a dose of 0.5 μg/mL PEDV, the number of PEDV (low) microcapsules was significantly higher than that in the PBS group; moreover, the number of PEDV (high) microcapsules was significantly higher than that in the PBS group. There was no significant difference between the microcapsule groups, inactivated group, and PBS group with regard to the specific antibodies. With the stimulating dose of 1 μg/mL PEDV, both PEDV microcapsule groups were significantly higher than the PBS group. The microcapsules and inactivated group were not significantly different from the PBS group. The lymphocyte proliferation results indicated that both 0.5 μg/mL and 1 μg/mL PEDV could stimulate T-cell proliferation with immune memory for the immune response of PEDV microcapsules.

Single-lymphocyte suspensions were isolated from animals after the last boost, plated as triplicates in a 96-well plate, and stimulated in vitro for 72 h with PEDV and con A as a positive control. T rate was analyzed with stimulation index: SI = (OD490 experiment group − OD490 culture media)/(OD490 control group − OD490 culture media). The differences between means were considered significant at * *p* < 0.05 and very significant at ** *p* < 0.01.

### 3.5. Microcapsules Enhanced the B Cell Differentiation

CD11b and CD11c are cell surface markers of dendritic cells which were detected by flow cytometric analysis (Figure 4A) from different groups of immunizations. The results showed that PBS, microcapsules, and inactivated PEDV stimulated approximately 2.2% of CD11b+ cells, and PEDV (low) microcapsules and PEDV (high) microcapsules stimulated CD11b+ cells. Conversely, the trend in CD11c+ cells gradually increased from the PBS group (4.3%) to PEDV (high) microcapsules (34.6%). The results of CD23 and B220 cell markers (Figure 4B) indicated that all B220+ cells were stimulated for differentiation in addition to the inactivated PEDV group, and the PBS and microcapsule groups reached 57.4% and 54.4%, respectively. Research on B220+ cells indicated that inactivated PEDV could better inhibit antibody production with oral immunization compared with other groups. Most importantly, the percentage of B220+ cells in both the PEDV (low) microcapsule and the PEDV (high) microcapsule groups was higher than that in the inactivated PEDV group. Nevertheless, B220+ and CD23+ cells were significantly increased by stimulation with PEDV; moreover, the percentages of these three groups were slightly different from those of the inactivated PEDV group (46.6%), the PEDV (low) microcapsules (45%) and the PEDV (high) microcapsules (47.7%).

### 3.6. Microcapsules Inhibit the Inflammation

To validate the results of inflammation after immunization, qRT-PCR was performed to analyze cytokine expression at the mRNA level (Figure 5). Compared to the samples from the different groups, our qRT-PCR results revealed significant differences in the expression of genes, such as IFN-γ, IL-4, IL-1, TNF-α, IL-17, IL-10, TGF-β, occluding, and ZO-1. The expression analysis was clustered by hierarchical clustering using the complete linkage algorithm and Pearson correlation metric in R for the heat map with GraphPad software. The heat map results indicated that the IFN-γ expression in the microcapsule group, PEDV (low) microcapsule group, and PEDV (high) microcapsule group was significantly higher than that in the inactivated group, and IFN-γ expression increased with increasing doses of PEDV. IL-4 expression showed a similar trend in the microcapsule group, PEDV (low) microcapsule group, and PEDV (high) microcapsule group, which stimulated the Th2 immune response. IL-4 expression in the inactivated group was different from IFN-γ expression, which indicated that the microcapsules could stimulate Th1 immune response. However, the two PEDV microcapsule groups exhibited different Th1 immune responses compared with the inactivated group.

The microcapsules could effectively inhibit the inflammation stimulated by the inactivated virus, which could decrease the expression of IL-1, TNF-a, and IL-17 in the microcapsule groups. The anti-inflammatory cytokines IL-10 and TGF-b were significantly increased in the microcapsules groups compared with the inactivated group. The relative tight junction genes also showed the same expression trend in various groups.

## 4. Discussion

The PEDV outbreak of 2013–2014 led to annual losses among worldwide swine producers. The loss of productivity from enteric diseases of PEDV in neonatal piglets costs swine producers millions of dollars [2,22]. Previous development of live and attenuated vaccines for another diarrheal virus of pigs, transmissible gastroenteritis virus (TGEV), provided insights into the mechanisms of mucosal immunity with IgA and piglet protection [5,23]. A previous study showed that the inactive PEDV was effective in stimulating the specific antibody of IgA in an immunized sow with attenuated vaccines protecting the piglets for lactogenic immunity [24]. Lactogenic immunity from pregnant sows is induced via the gut–mammary gland secretory IgA (sIgA) axis, which is also a promising and effective way to protect suckling piglets from PEDV infection [25]. Therefore, a successful PEDV vaccine must induce sufficient maternal or self-reproductive IgA antibodies [26,27]. However, the PEDV did not use the same route as the TGEV immune plan. The main reason is that the antibody titers of PEDV did not reach the titers of TGEV to boost a successful immune response. More research has focused on recombinant vaccines with different delivery vectors to protect antigens not digested by gastric acid [28]. However, the exposure dose of antigen was too low to stimulate specific antibodies, such as IgA and IgG. The antigen expressed by the recombinant vaccine was very exclusive. We explored a new form of possibility to deliver the virus into the gut, stimulating the specific mucosal immune response, which could directly neutralize the enterovirus at the site of infection.

Alginate microcapsules using ionic gelation represent a new and interesting approach to oral delivery systems for inactivating PEDV carriers. The microcapsules could efficiently capsulate the inactive PEDV antigen, and the whole virus particle was contained in the microcapsule. The microcapsules had good release profiles of the virus particles in saline solutions, such as PBS and normal saline, and the release rate was higher than 50% within three days. Studies have shown that the specific antibodies of IgA reach a peak in the second after immunization [29]. The most important feature was the microcapsules protecting the integrity of virus particles in hydrochloric acid, which prevents the virus particles from being released and subsequently digested in gastric acid [30]. Phage encapsulation and subsequent release kinetics revealed that the microcapsules possess pH-responsive characteristics with phage release triggered in an intestinal pH range suitable for therapeutic purposes [31]. A number of previous studies have used alginate as the main encapsulating agent either on its own or in combination with chitosan [32,33]. The chitosan–alginate microspheres effectively protected the virus in simulated gastric conditions, showing the remaining viral titers. The chitosan–alginate microspheres acted as an acid barrier in the simulated gastric conditions, helping the virus arrive at the M cell in the intestinal mucosal immune system. With storage at room temperature, the PEDV microcapsules displayed a remarkably low loss of antigen in the first three months, and they had good storage tolerance, thus maintaining the quality of the PEDV antigen. Tan reported that tocotrienols encapsulated in chitosan–alginate microcapsules have effective storage tolerance in the yogurt matrix [34].

PEDV initially attacks neonatal intestinal epithelial cells in piglets and the intestinal tract system, inducing diarrhea clinical signs, but systemic lymphoid organs cannot provide the effective antibody IgG to neutralize this virus in the neonatal pig. They mainly depend on the innate immunity of maternal antibodies. Thus, we chose microcapsules carrying inactive PEDV to stimulate the mucosal immune system of neonatal piglets. The specific antibodies against IgA and IgG showed that the microcapsules stimulated the mucosal and systemic immune systems producing specific antibodies, such as IgA and IgG [5]. In the first mouse immunization experiment, the microcapsule groups had higher levels of antibodies than the inactivated PEDV group. Most importantly, the immunization of microcapsules in specific antibody production was dose-dependent, and the levels of specific IgA and IgG in the high-dose microcapsule group were significantly higher than those in the low-dose microcapsule group. There was a close connection between the specific neutralization of antibody titers and the dose dependence of microcapsules arriving at the mucosal surface. The results of 50% neutralization with specific IgG and IgA were also found during microcapsule immunization, and PEDV (high) microcapsule groups had better neutralizing activity than PEDV (low) microcapsule groups. The specific antibody of IgA had a greater efficiency of neutralization than the specific antibody of IgG in the same group. The main reason was that chitosan–alginate microcapsules stimulate humoral immunity at the mucosal area, which is mainly mediated by IgA antibodies as the predominant immunoglobulin, and serum-derived IgG also contributes to immune defense [35]. Specific antibody experiments have indicated that microcapsules stimulate specific mucosal and systemic immunity in mice. However, the mice are not the infecting hosts of PEDV, which means there is just the possibility of an immune response in piglets. There are many differences in immunization and infection between mice and neonatal pigs.

Virus particle carriers have shown higher potential as oral delivery systems of proteins and peptides, which are taken up by M-cells of Peyer’s patches in the gut [35]. As seen in the qRT-PCR results, the microcapsules of PEDV, which have been reported to immunize with microcapsules, were associated with higher levels of IFN-γ production related to the T helper 1 (Th1)-type immune response [36]. In contrast, immunization with microcapsules without PEDV promoted IL-4 secretion related to the Th2-type immune response. IFN-γ production was higher than IL-4 production after PEDV microcapsule immunization. The cellular immune response (Th1) and humoral immune response (Th2) were disrupted, and the Th1 type immune response was predominant [6]. In particular, the high-dose PEDV microcapsules enhance IFN-γ expression, which means that the PEDV microcapsule immunization mediates the T helper 1 (Th1)-type immune response, and the chitosan–alginate plays an immunoregulatory role in extroversion to the cellular immune response. The mouse oral immunization of recombinant *Lactobacillus casei* expressing the Dendritic Cell-Targeting Peptide Fusing COE Protein of PEDV also mediated the Th1 immune response in piglets [37]. The cytokine response was also analyzed to compare the inflammation, cellular and humoral immunity in oral immunization with PEDV microcapsules. IL-1 and TNF-α were also used to analyze the safety and inflammation with immunization, which did not stimulate the inflammatory response when using the oral PEDV microcapsules. Many more studies have demonstrated the chitosan–alginate induction of the Th1-type immune response for oral vaccination in mice, and studies on the effects of chitosan have indicated that chitosan-fed farm animals showed higher weight gains but a lower incidence of disease than unfed animals [38,39,40].

DC-specific delivery has been considered a promising strategy for facilitating the efficient recognition, processing, and presentation of antigens by DCs, leading to enhanced antigen-specific immunity. The dendritic cell-targeted chitosan nanoparticles for nasal DNA immunization suggest that targeted pDNA delivery through a noninvasive intranasal route can be a strategy for designing low-dose vaccines [41]. Without adjuvant immunization, the increase in CD11c+ during vaccination promotes germinal center induction and robust humoral responses [42]. When injected, these alginate ‘vaccination nodes’ containing activated DCs attracted both host dendritic cells and a large number of T-cells to the injection sites in mice, while some of the inoculated DCs trafficked to the draining lymph nodes [43]. Compared with the inactivated PEDV group, chitosan–alginate groups, such as high-dose PEDV microcapsules and low-dose PEDV microcapsules, significantly stimulated the CD11c+ increase after the last immunization. Taking the advantage of CD11c+ in vivo DC targeting into consideration, chitosan–alginate could presumably induce in vivo DC maturation more robustly than inactivated PEDV. The chitosan–alginate could first effectively induce DC maturation of CD11c+ through the interaction with immunoregulation of chitosan–alginate in the gut, which enhanced the specific immune response, inducing subsequent humoral immunity with mucosal immunity and the systemic immune response.

## 5. Conclusions

The oral microencapsulation was packaged with different titers of inactivated PEDV, which consisted of alginate and chitosan arriving and presenting to the gut in mice. It induced specific humoral and mucosal immunity. Specific antibodies from immunized mice, such as IgA and IgG, neutralized PEDV in vitro. Oral immunization also stimulated immunologic memory in mice. Alginate and chitosan not only act as capsule wall materials but also enhance the viability of immune cells such as DCs and B cells with the function of adjuvants. To the best of our knowledge, this study of microencapsulation is the first to package enterovirus in the immunization of mice even though the host of PEDV is pigs. Therefore, the host animals, pigs, should be studied with inactivated and microencapsulated PEDV. All the data derived from this study can be an important reference point for further research in this area.

## Figures and Tables

**Figure 1 animals-13-00889-f001:**
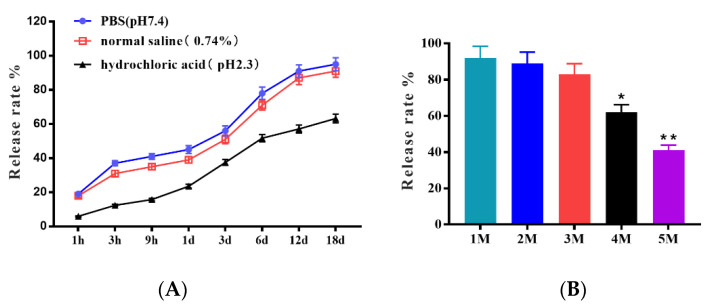
The stability of microcapsules shown by the release rate in (**A**) PBS, normal saline, hydrochloric acid, and release ability in different months (**B**). The differences between means were considered significant at * *p* < 0.05 and very significant at ** *p* < 0.01.

**Figure 2 animals-13-00889-f002:**
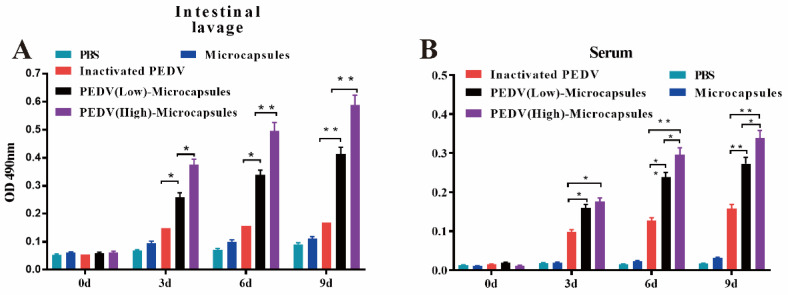
Enzyme-linked immunosorbent assay measures of antibodies IgA and IgG in feces (**A**) and serum (**B**) after the final immunity. The differences between means were considered significant at * *p* < 0.05 and very significant at ** *p* < 0.01.

**Figure 3 animals-13-00889-f003:**
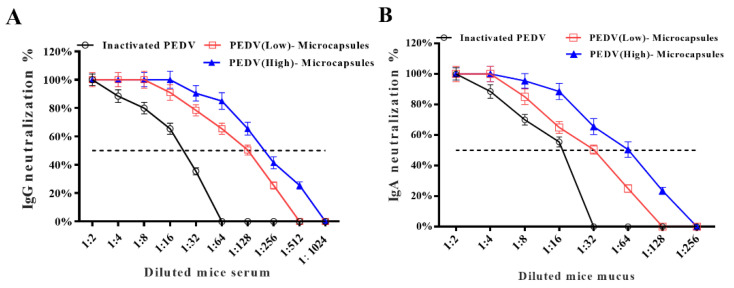
The results of antibody neutralization activity of IgG (**A**) and IgA (**B**). Both neutralization percentages of IgA and IgG in the intestinal mucosa and serum.

**Figure 4 animals-13-00889-f004:**
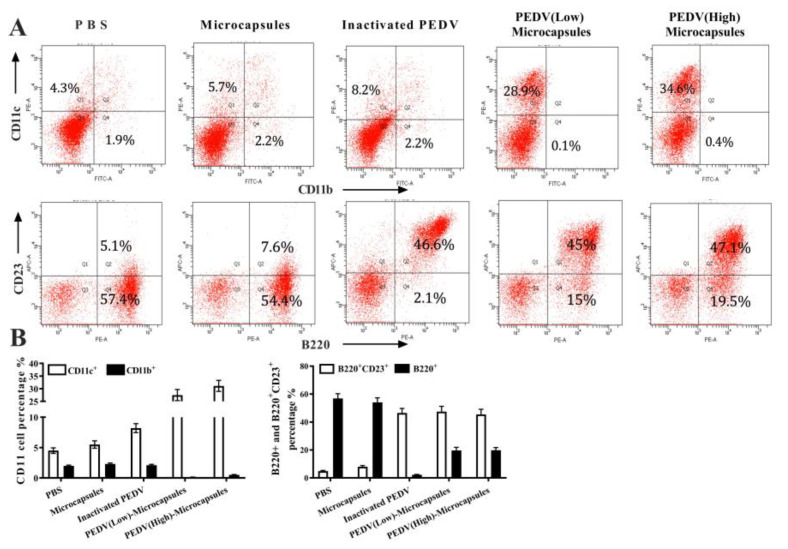
The flow cytometric analysis results for the percentage of CD11b+ and CD11c+ cell for immunization (**A**). (**B**) indicates the percentage of B220+ and CD23+ cell with the flowcytometric analysis.

**Figure 5 animals-13-00889-f005:**
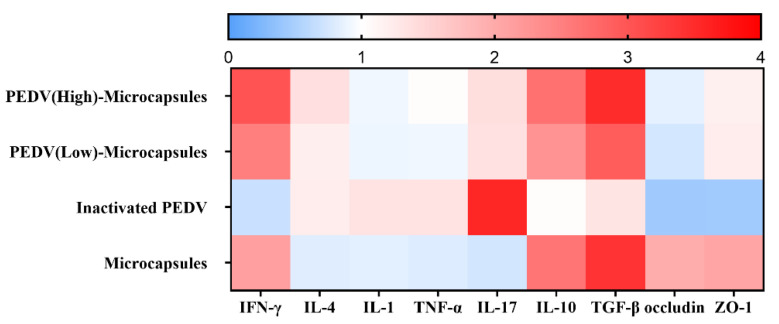
The expression analysis of genes by qRT-PCR analysis. For the expression analysis of genes, blue and red indicate decreased and increased expression, respectively. For qRT-PCR analysis of the expression of randomly selected novel genes from the immune and tight junction, data are presented as mean ± S.D. (n = 3).

**Table 1 animals-13-00889-t001:** Immunization regime of mice.

Groups (n)	Type of Vaccine	1st Vaccination		2nd Vaccination	
		Route	Dose	Route	Dose
PBS (10)	Negative control of PBS buffer	O/A	0.2 mL	O/A	0.2 mL
Microcapsules (10)	PBS encapsulated in alginate and chitosan	O/A	0.2 mL	O/A	0.2 mL
Inactivated PEDV(10)	Inactivated PEDV vaccine	O/A	0.2 mL	O/A	0.2 mL
PEDV (low) microcapsule (10)	PEDV vaccine encapsulated in alginate and chitosan with low titer virus (6 × 10^6^ PFU)	O/A	0.2 mL	O/A	0.2 mL
PEDV (high) microcapsule (10)	PEDV vaccine encapsulated in alginate and chitosan with high titer virus (6 × 10^7^ PFU)	O/A	0.2 mL	O/A	0.2 mL

n = number of mice; O/A = oral administration.

**Table 2 animals-13-00889-t002:** Lymphocyte proliferation index.

Groups	Stimulation Index Value	
	0.5 μg/mL PEDV	1 μg/mL PEDV
PBS(Control)	1.054 ± 0. 13	1.025 ± 0.11
Microcapsules	1.272 ± 0. 149	1.231 ± 0.156
Inactivated PEDV	1.349 ± 0.151	1.358 ± 0.166
PEDV (low) microcapsule	1.721 ± 0.171 *	2.529 ± 0.183 **
PEDV (high) microcapsule	1.987 ± 0.177 **	3.161 ± 0.191 **

## Data Availability

Once this manuscript is accepted, the data supporting the results of this study will be made publicly available in any publicly accessible repository.

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
