# Peer review of "The Oral Inactivated Porcine Epidemic Diarrhea Virus Presenting in the Intestine Induces Mucosal Immunity in Mice with Alginate–Chitosan Microcapsules"

_animals, 2023, doi:10.3390/ani13050889_

Round 1

Reviewer 1 Report (New Reviewer)

Dear Editor,

The manuscript entitles “The Oral Inactivative Porcine Epidemic Diarrhea Virus Present-2 ing in the Intestine Induces Mucosal Immunity with Alginate-3 Chitosan Microcapsules” was critically reviewed. In the present manuscript, the authors conducted an experimental study in mice evaluating the induction of mucosal immunity of orally administered inactivated PEDV, supposingly encapsulated with algenate-chitosan. The manuscript reports the induction of mucosal immunity using such method. However, the manuscript in its current format is far from consideration for publication. Several concerns are addressed, as following.

Concerns;
⁃ Language should be edited by native English speaker.
⁃ Abstract should be rewritten as it was not represented in what have performed.
⁃ The study was rather a duplication of what previously reports, please see PEDV and encapsulation topics in pubmed. More importantly, pigs have been used for evaluation.
⁃ PEDV causes a disease in pigs. Mice might not be a good candidate for studying pig mucosal immunity and oral vaccine in pigs.
⁃ Pig has long and complicated intestinal tract, unlike mice. Therefore, the control release study in mice might not be representative of what would happen in pigs.
⁃ It is strange that the authors can differentiate between IgA and IgA using virus neutralization (VN) assays.
⁃ It is also strange that why the authors did not use the same samples for both ELISA and VN assays.

Author Response

Respose to the Review Comments

We would like to thank you for your careful reading, helpful comments, and constructive suggestions, which has significantly improved the presentation of our manuscript. We have carefully considered all comments from the reviewers and revised our manuscript accordingly. The manuscript has also been double-checked, and the typos and grammar errors we found have been corrected. We are pleasure to receive new suggestion from the reviewers. We hope our revised manuscript can be accepted for publication. Thank you Very much for your review.

Best wishes,

Xinpeng Jiang

Concerns
⁃ Language should be edited by native English speaker.

Thank you for your professional suggestion. We have sent our manuscript for a English editing company. We will do our best in the correction of expression.
⁃ Abstract should be rewritten as it was not represented in what have performed.

Thank you for your professional suggestion. There was some mistake in the expression. We have rewritten the abstract, and added some sentences in the abstract. The correcting abstract have been marked with the yellow background. We hope the correcting abstract to promote the understanding for author.
The study was rather a duplication of what previously reports, please see PEDV and encapsulation topics in pubmed. More importantly, pigs have been used for evaluation.

There were some difference of our research comparing with the other studies in the pubmed, such as antigen, immunization route, wall material. In the PEDV studies, most of antigen were used recombined protein from the gene engineering. And there was no research with the inactivated PEDV or other coronavirus. The most important was that the combination between inactivated PEDV and oral route with the alginate-chitosan for the wall material. Our study focused on the oral inactivated PEDV, which was packed with the wall material for the alginate-chitosan. The alginate-chitosan have been studied for packing different protein, but only our PEDV and PRRSV in the pig virus immunization. The combination of alginate-chitosan and PRRSV was the

⁃ PEDV causes a disease in pigs. Mice might not be a good candidate for studying pig mucosal immunity and oral vaccine in pigs.

Thank you for your professional comment. The first steps, the mice was the model animal, which was used to analyze the safety evaluation, stimulating the specific immunizations of mucosal and humoral immunization and neutralization activity with the oral encapsulation. In the mice experiments, we make sure that the specific antibodies of IgA and IgG could neutralize the PEDV in the Vero cell. Secondly, when we make sure all these questions in the model animal of mice, we will immunize the piglet for studying pig mucosal and humoral immunity in host. Lastly, the immunized piglets would be challenged with the PEDV.

⁃ Pig has long and complicated intestinal tract, unlike mice. Therefore, the control release study in mice might not be representative of what would happen in pigs.

Thank you for your professional suggestion. Our team focused on the mucosal and humoral immunity for nearly twenty years. Even through the pig have a complicated intestinal microbe and mucosal immunity, we have built the method to test the specific antibodies of IgA in pig. We would study the oral encapsulation containing the PEDV with the release study in the piglet, which will be much more oral dose in piglet than the mice. We hope that the oral encapsulation could stimulate the specific immunization in the pig. The main problem was the concrete effects of oral immunization should be studied.  
⁃ It is strange that the authors can differentiate between IgA and IgA using virus neutralization (VN) assays.
⁃ It is also strange that why the authors did not use the same samples for both ELISA and VN assays.
Thank you for your professional comments. We answered these two similar questions together about IgA. We have added some detail about sampling in ELISA and VN. The The ELISA sample come from the extract supernatants of fecal samples, and the fecal samples contain too much bacterial toxin, which was bad to the cell in the virus neutralization assays. The VN sample come from intestinal lavage of mucus, most of fecal was taken out. and the mucus were filtered with 0. 45um filter membrane. We have rewritten some details about sampling in ELISA and VN.

Reviewer 2 Report (New Reviewer)

Just general comments,

1) The Porcine epidemic diarrhea virus (PEDV) is a Coronavirus, would it be possible to demonstrate how this virus causes infection in piglets, for example with a diagram or flowchart

2) What are the chances of PEDV to transmit directly/indirectly to humans? Are there any reports? How this study of yours can focus on this prospect and prevent future Coronavirus Outbreaks?

Author Response

Respose to the Review Comments

We would like to thank you for your careful reading, helpful comments, and constructive suggestions, which has significantly improved the presentation of our manuscript. We have carefully considered all comments from the reviewers and revised our manuscript accordingly. The manuscript has also been double-checked, and the typos and grammar errors we found have been corrected. We are pleasure to receive new suggestion from the reviewers. We hope our revised manuscript can be accepted for publication. Thank you Very much for your review.

Best wishes,

Xinpeng Jiang

1) The Porcine epidemic diarrhea virus (PEDV) is a Coronavirus, would it be possible to demonstrate how this virus causes infection in piglets, for example with a diagram or flowchart

Thank you for your professional suggestion. We have changed the abstract graph, and added some information about PEDV infection in piglet. We hope our change could satisfy your suggestion about the PEDV infection.   

2) What are the chances of PEDV to transmit directly/indirectly to humans? Are there any reports? How this study of yours can focus on this prospect and prevent future Coronavirus Outbreaks?

Right now, there was no evidence for the PEDV directly/indirectly transmitting to humans. But a novel HKU2-related bat coronavirus, swine acute diarrhoea syndrome coronavirus (SADS-CoV), was responsible for a large-scale outbreak of fatal disease in pigs. Both of PEDV and SADS-CoV caused severe intestinal diarrhea and death in fetal piglets, and there was no significant clinical signs and symptoms in infecting adult pigs. However, there was no research and reference on the PEDV infecting the human, even though the porcine intestinal coronavirus in the human. We could not deny the possibility of PEDV infecting the fetal infant in the future research. The SARS-CoV-2 and porcine intestinal coronavirus infect the epithelial cells in the lung or/and intestine, and the most of coronavirus would have the superinfection and persistent infection in the epithelial cells, which was main for the lower titers’ specific IgA from mucosal immunization in the surface of epithelial cells. Our team focused on the specific mucosal immunization against the coronavirus infection, which could also inhibit coronavirus entering into the epithelial cells.

Reviewer 3 Report (New Reviewer)

Dear authors,

The manuscript "The oral inactive porcine epidemic diarrhea virus..." has relevant results and an adequate methodology.

However, the manuscript presents some issues that should be evaluated by the authors.

1) The references are not adequate. For example, Ref 1 is not specific about the PEDV mortality rate. Ref 2 is not about systemic immunity in pigs. Ref 3 is extremely difficult to find. Ref 4 should be used at the end of the first paragraph and not where it is used (line 65). Finally, a careful review of the revisions used is lacking.

2) Adequate description of the methodology is missing. The authors must describe the method of obtaining and processing the mucus. They should describe how the release rate/month was made. The authors do not describe the method used for the filtration of serum and mucus in the neutralization experiments. The description of the statistical analysis must be redone. The cited reference does not bring additional information about the statistical analysis performed. For example, have the data been analyzed for normality?

3) In addition, tables 1 (line 150) and 2 (line 258) must be shown properly. Figures must be resized. Figures 3 and 4 should be enlarged and figure 5 reduced.

Author Response

Respose to the Review Comments

We would like to thank you for your careful reading, helpful comments, and constructive suggestions, which has significantly improved the presentation of our manuscript. We have carefully considered all comments from the reviewers and revised our manuscript accordingly. The manuscript has also been double-checked, and the typos and grammar errors we found have been corrected. We are pleasure to receive new suggestion from the reviewers. We hope our revised manuscript can be accepted for publication. Thank you Very much for your review.

Best wishes,

Xinpeng Jiang

1) The references are not adequate. For example, Ref 1 is not specific about the PEDV mortality rate. Ref 2 is not about systemic immunity in pigs. Ref 3 is extremely difficult to find. Ref 4 should be used at the end of the first paragraph and not where it is used (line 65). Finally, a careful review of the revisions used is lacking.

Thank you for your professional suggestion. The reference was not adequate, and we have corrected the reference. This reference introduced a part about Age-dependent resistance to PEDV infection.

The reference 2 and 3 were mistake and not adequate, we have changed the reference, and we supposed that the reference 2 and 3 could be unified for one review reference from Saif, L. J. in the Ohio State University. This reference has displayed the comparison about specific mucosal immunization and systemic immunization. The sows systemically immunized with inactivated TGEV vaccines had mainly IgG antibodies in serum and colostrum that declined rapidly in milk and provided little lactogenic immunity to piglets. Our discovery was focused on maternal vaccination strategies to induce mucosal passive immunity applicable to PEDV infection.

We have corrected this sentence for reference 4 into the end of the first paragraph, which will be much better for author to understand our research aim.

2) Adequate description of the methodology is missing. The authors must describe the method of obtaining and processing the mucus. They should describe how the release rate/month was made. The authors do not describe the method used for the filtration of serum and mucus in the neutralization experiments. The description of the statistical analysis must be redone. The cited reference does not bring additional information about the statistical analysis performed. For example, have the data been analyzed for normality?

Thank you for your professional suggestion. We have redone the statistical analysis in the release rate/month, we compared the first month release rate with another four months of release rate. We supposed that the release rate was main to prove the high stability in the normal storage environment. There was no standard about the best stability in the normal storage environment. We just want to indicate the quality of sodium alginate and chitosan in protecting the PEDV antigen.

All the samples in the neutralization assay much be filtered the microbe in the aseptic conditions. The detail and method for the filtration of serum and mucus were added in the PEDV neutralization assay part. The change part has been marked with yellow background. And we have also added some details about the statistical analysis in the experimental replication.

3) In addition, tables 1 (line 150) and 2 (line 258) must be shown properly. Figures must be resized. Figures 3 and 4 should be enlarged and figure 5 reduced.

Thank you for your professional comments. The formation of Table 1 and Table 2 have been corrected with the three-line table.

And the figures in our manuscript have been also reformatted as the reviewer’s requirements. The change part have been marked with yellow background.

This manuscript is a resubmission of an earlier submission. The following is a list of the peer review reports and author responses from that submission.

Round 1

Author Response

We would like to thank you for your careful reading, helpful comments, and constructive suggestions, which has significantly improved the presentation of our manuscript. We have carefully considered all comments from the reviewers and revised our manuscript accordingly. The manuscript has also been double-checked, and the typos and grammar errors we found have been corrected. In the following section, we summarize our responses to each comment from the reviewers. We hope that our responses have well addressed all concerns from the reviewers. Meanwhile, we are pleasure to receive new suggestion from the reviewers. We hope our revised manuscript can be accepted for publication. The manuscript with mark and respose to reviewr has been submit in the attachment. Thank you Very much for your review.

Best wishes,

Xinpeng Jiang

Reviewer 2 Report

The manuscript appear to present some interesting data from immunization of mice using encapsulated virus. Unfortunately, the merit of results and methods are hard to decide due to the "English language and style". The manuscript needs to be rewritten with an much improved English language before a judgement of the soundness can be done. 

Author Response

We would like to thank you for your careful reading, helpful comments, and constructive suggestions, which has significantly improved the presentation of our manuscript. We have carefully considered all comments from the reviewers and revised our manuscript accordingly. The manuscript has also been double-checked, and the typos and grammar errors we found have been corrected. We are pleasure to receive new suggestion from the reviewers. We hope our revised manuscript can be accepted for publication. Thank you Very much for your review.

Best wishes,

Xinpeng Jiang